# Plant Fibre Crafts Production, Trade and Income in Eswatini, Malawi and Zimbabwe

**Gladman Thondhlana** \***, Deepa Pullanikkatil** and **Charlie M. Shackleton**

Department of Environmental Science, Rhodes University, Grahamstown 6140, South Africa;
d.pullanikkatil@gmail.com (D.P.); c.shackleton@ru.ac.za (C.M.S.)
**\*** Correspondence: g.thondhlana@ru.ac.za

**Abstract:** The production of plant fibre products is considered a promising pathway for contributing to people's livelihoods particularly in developing countries, where economic options might be limited. However, there are limited comparative studies across countries on plant fibre products, making it difficult to examine how local and broader biophysical, socioeconomic, cultural and policy contexts influence craft production patterns in terms of primary plant resources used, products made and contributions to livelihoods. Using household surveys for data collection, this paper presents findings from a comparative analysis of plant fibre craft production and income in three southern African countries, Eswatini, Malawi and Zimbabwe. Although there was commonality in terms of the constraints experienced across the three countries, there were pronounced differences in the types and quantity of products and income between and within countries. The average gross monthly income from craft sales was modest and of the same order of magnitude across the three countries but 50% higher in Zimbabwe (US$75 ± 135) than in Eswatini (US$56 ± 71) and Malawi (US$48 ± 168). High craft income was associated with long experience in craft production, quantity of craft products and access to bulk buyers while old age, more income sources, high education level and bigger households yielded low craft income. Although craft income tended to be low, the economic contexts in these countries characterised by high levels of poverty, craft income represents an important livelihood source. Implications for policy interventions are discussed.

**Keywords:** plant fibre; crafts production; income; heterogeneity

## 1. Introduction

The production of plant fibre crafts such as mats, baskets, brooms and ropes is a key livelihood activity associated with the socioeconomic and cultural life of environmental resources-dependent communities globally [1–6]. Many of these craft products have several functional and decorative roles at the household and community level and are also popular mementoes and collectables for tourists [7]. The importance of plant fibre products is seen in the number of people involved in production and the incomes generated from trade [8]. In many places the production and trading of plant fibre products provide important cash incomes for many rural households [2,4,7,9–14]. The relative contribution of income from plant fibre products to households varies from small, supplementary contributions to substantial shares.

The production of plant fibre crafts is of particular importance in developing countries, where economic options may be limited. For example, evidence suggests the importance of handicraft to the livelihoods and well-being of many households in Brazil [4], India [15], and Pakistan [6,16]. Particularly, in southern Africa, craft production is the mainstay of many household economies [1,12,13,17]. In Mozambique, Ref. [18] showed that communities use a wide range of plant species to produce handicrafts for subsistence use. In South Africa, some studies show the mean contribution of plant

fibre crafts to total household income ranges from 35% to 51% [13,19]. In absolute terms, mean annual income from trade of plant fibre products has been found to be as low as US$144 [12] to as high as US$1077 [19] per crafter in the Eastern Cape and Limpopo provinces of South Africa, respectively.

In addition to regular cash income generation, the trade of crafts can act as a safety net during times of economic hardships, providing a livelihood cushion to many poor households [11,20,21]. For example, small enterprise activities using natural resources provide a crucial safety net for HIV/AIDS-affected households in South Africa [19] and the broader region [22]. Various studies show that such activities are an important source of livelihood, especially for the marginalised and vulnerable members of rural communities such as the disabled, elderly and poor women, who often have limited options for cash income generation [17,21,23,24]. For instance, Ref. [12] found that the majority of the traders of hand brushes, made from fronds of wild palms, were women over the age of 50 years who had begun trading in response to hardship and the need for cash income.

Beyond these direct-use values, the production of plant fibre crafts may also increase the sense of dignity and well-being among marginalised rural women and give them a chance to redefine their role in society [25]. In demonstrating the cultural significance of the practice of craft production and trade, Ref. [26] shows that communities travel greater distances to harvest plant species of traditional value, due to low preference of the substitutes for such natural products. Similarly, Ref. [27] revealed that 70% of urban buyers of traditional plant fibre brooms did so for cultural purposes rather than utilitarian ones. Many plant fibre crafts have cultural significances such as being used for weddings or funerals.

Many studies, however, have found that the optimum contribution from crafts is often constrained by household and contextual factors, including poor and changing markets, resources located far away and lack of capital, skills and transport [12,13,24,28]. These studies suggest that overcoming these barriers through practical and policy interventions could enhance the contribution of crafts to crafters' livelihoods, which can reduce the burden on local government authorities to provide livelihood security [19]. Cunningham and Terry [1] noted that commercialisation of baskets positively affected people's livelihoods, although there was often a need for improved resource management and product quality to optimise the benefits especially in relation to urban and international markets. Indeed, the increasing integration of crafts into tourist and export markets places crafters at risk of fluctuating currency exchange rates, changing fads and designs, competition from other sources and the need for greater resource supply to meet market demands [1,28], meaning that they need to be more responsive and adaptable. Consequently, the production of plant fibre crafts has attracted global attention, especially for the role they play in potentially lifting the poor out of poverty or ensuring livelihood security for vulnerable households.

The broader literature on wild resource dependence has grown exponentially in the last decade [29], and production of plant fibre crafts is a promising pathway for contributing to people's livelihoods. However, to the best of our knowledge, we are unaware of any comparative studies across countries on plant fibre products, other than that of [1]. Arguably, most studies on NTFPs focus on wild foods, medicine, building material and energy [29,30]. Despite increasing studies on plant fibre products, most are based on one or two villages or regions [6,12,15,31], while some are based on personal stories of individuals [8]. Moreover, in southern Africa, most studies are from South Africa with relatively few from other countries in the region, despite a rich heritage of plant fibre products production and use [1]. Consequently, it is not possible to examine how local and broader biophysical, socioeconomic, cultural and policy contexts influence patterns in terms of primary plant materials used, products made and contribution to livelihoods. Moreover, attempts to do so from existing studies are fraught with methodological differences [29]. Consequently, comparative studies are useful and necessary to tease out such contextual specifics and importantly, identify appropriate and supportive policies. Therefore, we argue that more work is needed to fully understand how plant fibre craft production, trade and contribution vary and that the insights comparative research could bring will provide a much stronger basis for strategies to support local level craft production and trade.

Against this background, the main aim of this study was to examine the extent, dependence and livelihood contribution of plant fibre crafts in three southern African countries, Eswatini, Malawi and Zimbabwe, as a basis for identifying strategies for optimising benefits from production and trade of plant fibre products. Within the three countries, we addressed the following questions: (i) what are the reasons crafters engage in plant fibre craft production and trade, (ii) what are the processes of craft production and the range of products produced, (iii) what is the contribution of craft trade to crafters' income and (iv) what household and contextual factors (including challenges) determine income from crafts. Though the use of plant fibre products for cultural purposes is important, the scope of this study is limited to the production and livelihood uses of fibre products.

## 2. Study Area

The study took place between October and November 2016 in three southern African countries, Eswatini, Malawi and Zimbabwe. Within each county, at least three study sites across varying topography, rainfall, ethnic groups, and market access were selected. The selection of different sites was based on ecological zones and proximity to markets, which together may influence the type of plant fibres used and the magnitude of the contribution of plant fibre crafts trade to household incomes. In Malawi, seven sites in the central and southern regions were selected, while in Eswatini two sites were chosen in each of the Highveld, Middleveld and Lowveld regions. In Zimbabwe, two sites in the Lowveld and one in the Eastern Highlands were selected (Figure 1). The sites chosen included mountainous areas, hills, forests, rural farmlands, periurban settings, as well as cities (Table 1). In all the three countries, most crafters had free access to plant fibre within their localities except for a few crafters in Zimbabwe who either bought from other areas or harvested from neighbouring Mozambique. Seven languages were spoken across all the study sites in the three countries.

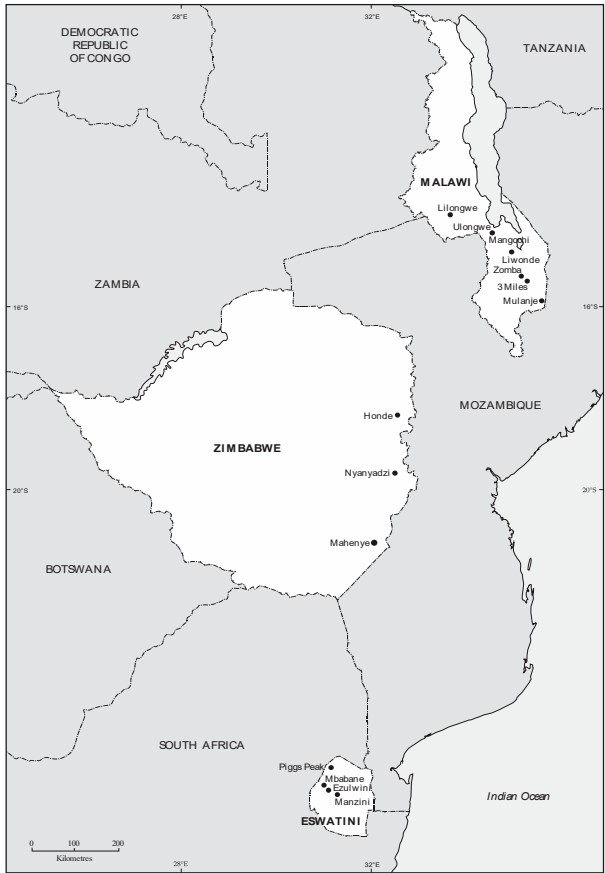

**Figure 1.** The location of the study sites in Eswatini, Malawi and Zimbabwe.

**Table 1.** Selection attributes of the different study site.

| Country | Study Sites | Languages | Landscapes and Use | MAR (mm) |
|---|---|---|---|---|
| Eswatini | Piggs Peak (Highveld) | Siswati | Forests, mountains | 2500 |
| | Mbabane (Highveld) | Siswati | Mountains, hills, urban areas | 2500 |
| | Ezulwini (transition line from Highveld to Middleveld) | Siswati | Valley, urban areas | 750–500 |
| | Manzini (central region covering Highveld, Middleveld and Lowveld) | Siswati | Hills, lowlands, urban areas | 750–500 |
| Malawi | Lilongwe (central region) | Chichewa, Chiyao | Hills with extensive agriculture and savannah, urban | 800–1000 |
| | Ulongwe, Mangochi, Liwonde (southern region) | Chichewa, Chiyao, Lomwe | Lowlands with extensive agriculture and savanna | 800–1000 |
| | Zomba, 3 Miles (southern region) | Chichewa, Chiyao, Lomwe | Mountains and hills with forests and intensive agriculture, urban areas | 1000–1200 |
| | Mulanje (southern region) | Chichewa, Chiyao, Lomwe (Ngoni) | Mountains with forests and intensive agriculture | 2000–2200 |
| Zimbabwe | Mahenye (southeastern Lowveld) | Shangani/Shona | Lowlands, rural area situated further from main road | 400–500 |
| | Nyanyadzi (Lowveld) | Shona | Lowlands, rural area along a major road | <650 |
| | Honde Valley (Eastern Highlands) | Shona | Highlands, rural area | 1200 |

The three countries are generally characterised by high poverty levels and high unemployment rates, with a substantial proportion of the population dependent on land- and farming-based livelihoods, although with growing rural to urban migration. Wild resources, particularly nontimber forest products (NTFPs), are important sources of livelihoods, for both household use and cash income in all the three countries (e.g., [32–34]). Further, tourism is vitally important to the economies of all three countries.

*Data Collection and Analysis*

Questionnaires were administered in the local languages of crafters in the different sites. In areas where researchers were not familiar with the local language, translators were employed. Crafters were approached either at their homes (via a snowball approach) or place of trading (selling points or along the roads). Traditional Authorities helped identify a few producers after which snowball sampling was used to identify households that engaged in production and trade of plant fibre crafts.

In total, 343 plant fibre crafters from Eswatini (100), Malawi (100) and Zimbabwe (143) were interviewed. The questionnaire was designed to get information on (i) the socioeconomic profile of the crafters and their households and (ii) production and trade of all different types of plant fibre crafts produced. Sociodemographic information obtained include age, gender and education level of the respondent, household size and employment status and sources of income.

Information regarding the crafters' length of involvement in craft production and trade and reasons for engagement in craft production were elicited. Respondents were also asked about their perceptions on demand for plant fibre crafts, trends in abundance of plant resource supply and number of people operating in the trade over the last 7 years, as well as the challenges they faced.

The questionnaire also captured information on the primary plant species used in craft production, the range, types and quantities of crafts produced and traded and the prices if traded. The crafters were also invited to comment on whether or not they thought their crafts were distinct from other regions in their respective country. Local names of species used and why they were preferred and when and how they are harvested were also documented. The harvesting and production process and methods were noted.

The survey data were entered into an Excel spreadsheet. We estimated gross monthly income based on the crafters' recall of sales in good months and bad months (including number of good and bad months) per year and the selling price per unit. At the time of the study, the Malawian Kwacha-dollar and Eswatini Lilangeni-dollar exchange rates were USD1 = MWK719 and USD1 = SZL14.5, respectively. Zimbabwe, traded in the US dollars as it had stopped using its local currency due to hyperinflation.

STATISTICA Version 13 (TIBSCO Software Inc., Palo Alto, California, USA) was used for statistical analysis. Descriptive statistics were used, where relevant, to show the distribution of data (e.g., gross monthly income from the trade of crafts, reasons for trading, perceptions on number of traders and customers, trends in availability of plant resources used for craft production and constraints to trade) using proportions either in text or graphical summaries. Continuous data (age, household size and incomes) were summarised in the form of means and standard deviations. Differences between means were tested using *t*-tests and Mann–Whitney U tests for parametric and nonparametric data, respectively. A one-way ANOVA and Tukey HSD tests were performed to test if there were significant differences in selected continuous variables between countries and sites within countries. A generalised linear regression model was used to test the predictors of craft income because it allows modelling of correlated and non-normally distributed data with flexible accommodation of covariates [35]. To test for the robustness of regression coefficients (i.e., consistency of *p*-values) [36], we used the restricted or residual maximum likelihood approach. Independent variables considered in the regression model were either household- or site-level predictors. Household-level variables included indicators of human capital (e.g., household size, age, education and gender of household head and employment status) and contextual variables that might influence income from crafts including variety and number of crafts produced, access to a formal market (where sellers can publicly advertise their prices and locations and are subject to local taxes and regulations) and variety of buyers. Ethical clearance from Rhodes University was granted, and permission from Traditional Authorities at each site was sought before the study began.

## 3. Results

### 3.1. Sociodemographic Data of the Respondents and Households

The sociodemographic data provided by respondents was generally comparable to the official national demographic information of the respective countries, but was different between countries for some attributes. Nearly all interviewed respondents in Eswatini were females, but females were less than half and one-third of the interviewed respondents in Malawi and Zimbabwe, respectively (Table 2). The average age (± SD) of the respondents was slightly, but significantly higher in Eswatini (49 ± 14) than in Malawi (42 ± 14) and Zimbabwe (42 ± 3). Most respondents had low education levels. Though the average number of years spent in school was significantly higher in Eswatini and Zimbabwe than in Malawi, it was 7 years or below, suggesting a considerable number of household heads had achieved only primary level education. In all three countries, 10–20% of respondents had no formal education, but Zimbabwe had the lowest proportion. Average household size was similar across the three countries. Many households owned houses constructed from locally acquired materials, including wood, clay and mud, cow dung and thatch grass, or a combination of modern houses (brick, cement floors) and traditional ones. In all three, countries the average number of rooms per dwelling was three. Less than half of the households owned a television set and radio except for Eswatini where some owned refrigerators, a bicycle and car/donkey cart.

Although variable between countries, most households stated diverse sources of income, including crop and livestock production, wages, social welfare grants, remittances and other self-employment activities. Notably, crop production was a common income source for most households in Zimbabwe (88%) and Malawi (65%). Livestock production was also reported as an important income source in all the three countries but more so in Eswatini (89%) and Zimbabwe (86%) than in Malawi (49%). Many livestock-owning households had more goats than cattle, except for Eswatini. Other livestock

types, i.e., pigs, horses, ducks and donkeys were reported by only a few households. Self-employment, including small business enterprises and petty trade of grocery items, clothes and other household items, were also commonly cited income sources. In Malawi, the respondents cited selling of brooms, baskets and fish, and transport provision as important sources of income, whereas in Zimbabwe, temporary jobs like brickwork, dress-making, plastering, carpentry and fishing were key livelihood activities.

**Table 2.** The respondent and household (hh) socioeconomic profile of the sampled populations in the three countries.

| Attribute | Eswatini | Malawi | Zimbabwe | ANOVA (F) |
|---|---|---|---|---|
| Number of respondents | 100 | 100 | 143 | – |
| % female | 99 | 26 | 41 | – |
| Age (years) | 49 ± 14 [b] | 42 ± 14 [a] | 42 ± 3 [a] | F = 8.212 ** |
| Education (years) | 7 ± 4 [b] | 5 ± 4 [a] | 7 ± 3 [b] | F = 5.522 * |
| % of respondents with no education | 15 | 18 | 9 | |
| Household size (persons) | 7 ± 4 | 6 ± 4 | 7 ± 3 | F = 2.130 |
| Number of children | 4 ± 3 | 3 ± 2 | 4 ± 2 | – |
| Marital status (%): | 23 | 9 | 7 | |
| Single | 63 | 84 | 84 | |
| Married | 14 | 6 | 9 | |
| Widowed/divorced | | | | |
| Average number of rooms per hh | 3 ± 2 | 3 ± 1 | 3 ± 1 | F = 0.810 |
| % of hh with television | 31 | 11 | 8 | |
| % of hh with radio | 66 | 51 | 36 | |
| % of hh with fridge | 28 | 5 | 1 | |
| % of hh with bicycle | - | 56 | 35 | |
| % of hh with motorcycle/car/donkey cart | 2 | 9 | 10 | |
| Other sources of income other than from trade: | 3 | 6 | - | |
| Wages | 26 | - | - | |
| State pension | 16 | 65 | 88 | |
| Crops | 89 (43) | 49 (25) | 86 (69) | |
| Livestock (excluding chickens) | 14 | - | 11 | |
| Remittances | 9 | 60 | 19 | |
| Other self-employment | | | | |
| % of hh with cattle | 31 | 4 | 28 | |
| % of hh with goats | 22 | 28 | 62 | |
| % of hh with chickens | 82 | 45 | 76 | |
| Average years in trade | 14 ± 12 [b] | 17 ± 11 [a] | 19 ± 12 [a] | F = 3.49 * |
| Reasons for entering trade (%): | 36 | 41 | 29 | |
| Poverty/unemployment/cash income | 7 | 4 | 1 | |
| Interested in crafts | 16 | 4 | 9 | |
| Low education level | 15 | 23 | 33 | |
| Only job known/skilled | 2 | 4 | 46 | |
| Easy availability of raw materials | 13 | 16 | 10 | |
| Low entry cost | 2 | 8 | 12 | |
| Societal influence | | | | |

**, * indicate 1% and 5% level of significance respectively; Letters [a], [b] indicate significant differences in means between wealth groups ($p < 0.05$).

### 3.2. Reasons for Production and Trade of Plant Fibre Crafts

On average, the respondents had been involved in craft production and trade for 14 years or longer, ranging from 14 ± 12 in Eswatini, 17 ± 11 in Malawi to 19 ± 12 in Zimbabwe (Table 1). In all the three countries, more than a third of the respondents had been involved in fibre craft production and trade for at least 20 years (Table 1). However, new entrants with 5 years or less in the production and trade of crafts constituted 11% (Zimbabwe) to 26% (Eswatini) of the sample. The most stated reasons for entering craft production and trade was the "need for cash income generation" and unemployment, stated by 41% and 36% of the respondents in Malawi and Eswatini, respectively. In comparison, in Zimbabwe, the availability of plant fibre resources was the most common reason (46%). Other reasons stated include interest in craft production, low education levels, lack of other livelihood skills, low entry cost and societal influence, with the proportion of varying between countries. With regards to societal influence, the respondents reported that craft production and trade was part of their family tradition and culture and that they had learnt craft production skills at a young age from their parents, relatives, neighbours and friends. Other reasons mentioned by the respondents for entering craft production include convenience, an opportunity to work from home, supplementing income, loss of a breadwinner (husband), stop-gap measure, social role of women, to keep oneself busy and inability to seek other jobs due to disability. In Malawi, 64% of the respondents said that they acquired craft-making skills from their relatives (especially mothers), friends (17%) and the remaining proportion learnt from school (9%) or taught themselves (10%). A considerable proportion of the respondents from Eswatini reported that they learnt how to make crafts from family members (40%) and other women (38%), and the remainder either taught themselves or benefitted from training programmes run by schools or nongovernmental organisations. In Zimbabwe, most respondents stated that they learnt craft production from family members (parents, husband, brother, grandmothers, etc.) and the remaining proportion from friends, neighbours and school.

### 3.3. Resource Collection, Treatment and Production of Crafts

The plant species required for craft production were mainly harvested locally in the three countries but crafters sometimes bought resources due to scarcity or time considerations. There were both variations and similarities in the plant types and parts used between and within countries. In Eswatini, products were mainly produced from different grass and sedge species, including *Digitaria swazilandensis* (Luthindzi), *Phragmites australis* (Lukhasi), *Cyperus latifolius* (Likhwane), *Juncus kraussii* (Incema), *Miscanthus capensis* (Umtsala) and *Cyperus articulatus* (Incobozi), except for a few products made from gourds, corn cobs and *Agave sisalana* (sisal). Some products were decorated using natural and chemical dyes. Harvesting (cutting) of Lukhasi in wetlands for making brooms and mats took about 9 h per trip. Treatment (drying, wetting and dying of grass) varied by product type. Drying of grasses typically took a day on sunny days or longer on cloudy days across all the sites. Twisting of grass took about 30 min to 1 h for flower and decorative pots and mats and up to a day for chicken coops and sleeping mats. Production time of grass-based products varied from 2 h for brooms to a day (about 8 h) for mats depending on specific product types and sizes. Collecting and processing (cutting, removal of pulp and drying of the shell) of butternut gourds and production of calabashes took approximately 2 days. Harvesting of sisal in the mountains typically took up to 4 h. Treatment of sisal involved scraping to make threads which typically took between 1 and several hours depending on the quantity and type of products made (beads, jewellery and accessories). Production of products involves rolling sisal threads on one's thighs, and it takes a day or more. The time taken to thread the treated and rolled sisal to make sisal beads for jewellery could be up to 2 h for big beads or less for small beads.

In Malawi, a variety of plants were used for craft production; grasses, bamboo, cane, mwanga and jacaranda seeds, mlaza (palm leaves), chawa and sisiri grass. Resource collection, treatment and production time varied by plant species used and quantity and type of products. Harvesting trips for bamboo from the forest took about 5 h to a day while treatment (soaking and cutting to size) took a

few hours and making the furniture ranged from 12 h to 3 days depending on quantity. Harvesting of cane, including removing the outer green cover and splitting took about 10 h while treatment (drying) of grass and weaving of baskets took between 3 and 5 h. Palm leaves were purchased from harvesters, and treatment (including cutting to required size and drying) of the palm leaves and weaving took 1 and up to 2 days, respectively. Collection (uprooting) of sisiri grass from the forests took about 6 h to fill up a 50 kg bag. Treatment stages involved leaving the bags in the sun for 7 days and spreading the "browned" grass in the sun for complete dryness for a single day. The time required to weave depended on product type and size and ranged from a few hours to a few days. Collection of grasses for broom production involved cutting it to size, drying and tying the grass to make brooms, which took anything from a single day to 2 days. The time required for collection of treatment of seeds and ornaments production was relatively short. Seeds were boiled and soaked in cooking oil for about 10–30 and 5 min, respectively, and production of a single necklace from treated seeds took about 45 min. The results show a lot of variation based on site, plant resources used and products.

In Zimbabwe, plant resources used for crafts varied by site, mainly, *Adansonia digitata* (African baobab) in Nyanyadzi, reeds in Mahenye and bamboo in Honde. Other plants such as *Agave sisalana* (sisal) and *Hyphaene petersiana* (palm leaves) were used at varying levels at the different sites. Harvesting of baobab bark took about a day, followed by treatment (splitting, soaking and dying), which typically took 1–2 days. Craft production of products (knitting and weaving), usually took between a few hours to 2 days depending on type and size of the products. Collection, treatment and production of crafts from sisal and palm leaves were similar to the processes followed in Eswatini, except that these resources were no longer readily available and were purchased in many cases. Harvesting (cutting), spitting and drying of bamboo and weaving took up to 4 days, but the actual weaving of individual products took between 30 min and 4 hours depending on size, design and type. Harvesting, treatment and weaving of reeds generally took longer than other plants. Harvesting (cutting), splitting, drying, threading, soaking and weaving mats took up to 6 h, at least twice the time taken in Honde (2 h) and Nyanyadzi (3 h).

### 3.4. Variety and Nature of Products and Perceptions on Resource Availability

Various types of plant fibre crafts were produced in the three countries including different types and sizes of baskets, mats, ornaments, toys, wall hangers and tables (Figure 2 and Table 3). Of all, the crafts, baskets, brooms, mats, hats and ornaments were commonly produced. The products were made from different plants and plant parts including grass, cane, reeds, seeds, palm leaves, sisal fibres, bamboo and baobab bark. In Eswatini and Malawi, there were no marked differences in plant types used by site but some species such as *Cyperus latifolius* (Likhwane) were available in wetlands, while others were available in mountains. In Zimbabwe, the plants used for craft production differed by ecological region; bamboo was dominant in the highlands (wet and cool regions), whereas baobab fibre, wild date palm and reeds were used in the Lowveld areas (dry and hot).

Craft producers mainly used locally available wild natural resources or purchased them in cases where they were not readily available. Most of the craft producers in Eswatini (70%) and Zimbabwe (75%) felt that the abundance of natural resources they required had decreased over the last 7 years (Table 4). The commonly stated explanations for the decline were (i) too many harvesters, (ii) droughts/climate change and (iii) wild fires. In Malawi, respondents stated that deforestation from land clearing for farmland or property development was a key driver of resource decline, whereas in Zimbabwe, the crafters mentioned competition with livestock and wildlife (mainly elephants) in particular areas. Those who felt it had remained the same (in Zimbabwe) said that some plants, like the baobab tree, recovered quickly. More respondents in Zimbabwe than in Eswatini and Malawi felt that the variety and nature of products had changed.

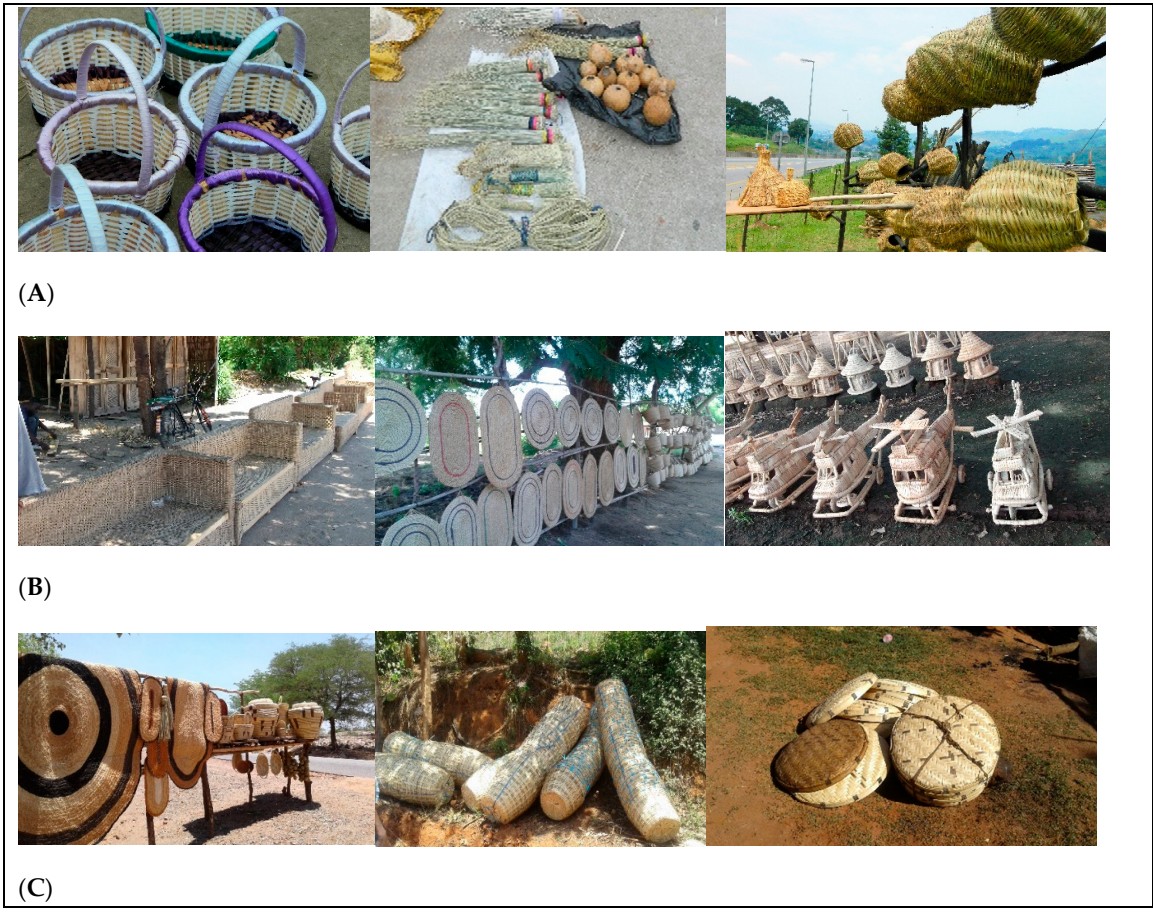

**Figure 2.** Photos of plant fibre products: (**A**) bamboo baskets; calabash from pumpkin and brooms and chicken poops in Eswatini, (**B**) cane furniture; floor mats and baskets and palm fibre toys in Malawi and (**C**) baobab fibre mats; and palm and bamboo baskets in Zimbabwe. (Photo credits: Deepa Pullanikkatil (**A**,**B**) and Gladman Thondhlana (**C**).

**Table 3.** Plant fibre crafts produced and plants used in the three countries.

| Country | Product(s) | Plant Forms, Part Used |
|---|---|---|
| Eswatini | Brooms, chicken coops, baskets, floor mats, coasters and table mats | Grasses and sisal |
| | Calabash | Gourds |
| | Decorative flowers and pots | Palm leaves, pine cones and grasses |
| | Ornaments | Sisal inserted into steel and silver jewellery |
| | Curtains | Grasses, corn cobs |
| | Basotho hat | Grasses |
| Malawi | Brooms, hats, baskets, sleeping mats and door mats | Grasses |
| | Furniture, caskets, floor mats, winnowers, baskets and decorative wall pieces, fish traps, bottle covers, lamp shades | Cane and bamboo |
| | Ornaments (necklaces/bracelets/earrings) | Seeds, carvings made from bark of tree |
| | Toys (toy house, aeroplanes, cars) and hats | Palm leaves |

**Table 3.** *Cont.*

| Country | Product(s) | Plant Forms, Part Used |
|---|---|---|
| Zimbabwe | Baskets (food, washing, pets, flower, fruit, winnowing, shopping) | Bamboo, palm leaves |
| | Brooms | Baobab bark, palm leaves |
| | Mats (door, floor) | Baobab bark, reeds |
| | Chicken coops | Bamboo |
| | Tables and chairs | Bamboo |
| | Doors | Reeds |
| | Handbags | Baobab bark |
| | Fish traps | Bamboo |
| | Hats | Baobab bark, palm leaves |
| | Spoons | Bamboo |
| | Shelves | Palm leaves |
| | Trays | Bamboo, palm leaves |
| | Wall hangers | Bamboo |
| | Lamp shades | Bamboo |

**Table 4.** Traders' perceptions on trends in resource availability, variety and nature of crafts, number of traders and customers over the previous 7 years.

| Attribute | Direction of Change | Country | | |
|---|---|---|---|---|
| | | Eswatini | Malawi | Zimbabwe |
| Abundance of natural resources | Improved | 17 | 8 | 4 |
| | Decreased | 70 | 48 | 75 |
| | No change | 2 | 44 | 21 |
| | Do not know | 11 | – | – |
| Variety and nature of products | Increased | 35 | 45 | 51 |
| | Decreased | 15 | 19 | 42 |
| | No change | 35 | 36 | 7 |
| | Do not know | 15 | – | – |
| Number of customers | Increased | 33 | 52 | 7 |
| | Decreased | 23 | 29 | 90 |
| | No change | 37 | 10 | 3 |
| | Do not know | 7 | – | – |
| Number of traders | Increased | 52 | 55 | 39 |
| | Decreased | 12 | 18 | 56 |
| | No change | 29 | 27 | 7 |
| | Do not know | 7 | – | – |

In all the three countries, a sizeable proportion of the respondents said that they had changed the variety and nature of their craft products in the past years, though more crafters in Zimbabwe (51%) than in Eswatini (35%) and Malawi (45%) said so. Analysis of the number of product types showed significant differences between countries with crafters in Zimbabwe reporting more product types (2.3 ± 1.5) than in Eswatini (1.6 ± 1) and Malawi (1.8 ± 1.3) (F = 8.93, $p < 0.001$). The quantity of products was also significantly different between the countries, with crafters in Malawi producing more products per month (216 ± 342) than in Eswatini (51 ± 73) and Zimbabwe (78 ± 101) (F = 17.73, $p < 0.001$). Further analysis by sites within countries shows that number of product types were comparable across sites in Eswatini (highveld, 1.4 ± 0.6; middleveld, 1.6 ± 1.1). Although crafters in the highveld sites of Eswatini produced more products (74 ± 109) than those in the middleveld (46 ± 63), the difference was not significant (Z = 0.46, $p > 0.05$). Similarly, in Malawi, there were no significant differences in the number of product types between medium rainfall sites (800–1000 mm) (1.8 ± 1.1)

and high rainfall sites (>1200 mm) (1.9 ± 1.6) (Z = 0.60, *p* > 0.05) but there were more products per crafter in high rainfall sites (358 ± 395) than in medium rainfall sites (162 ± 306) (Z = 2.41, *p* < 0.05). In Zimbabwe, the number of product types differed significantly by ecological region, with crafters at the humid site (Honde), producing significantly more product types (3.2 ± 1.1) than those in the dry lowveld sites, Nyanyadzi (2.5 ± 1.1) and Mahenye (1.2 ± 0.6) (F = 28.29, *p* < 0.001). Similarly, Honde showed a significantly higher mean number of craft products produced per month (156 ± 140) than in Nyanyadzi (28 ± 36) and Mahenye (49 ± 33) (F = 19.82, *p* < 0.001).

### 3.5. Perceptions on Customers and Craft Trade

In Eswatini, there was little agreement regarding the trend in customer numbers, with more respondents perceiving there was no change than those who thought there was an increase or decrease (Table 4). More than half (52%) of the respondents in Malawi stated that the number of customers had increased over the past decade, while 29% thought it had decreased. In both Eswatini and Malawi, respondents who stated customers had increased cited (i) better designs, (ii) exposure to markets through support from NGOs, (iii) publicity and (iv) good customer care, as reasons behind the increase. Those who perceived a decrease in numbers of customers cited (i) an increase in the number of traders and hence more competition and (ii) an increase in number of people making their own products for household use. A substantial proportion (90%) of the respondents in Zimbabwe perceived the number of customers had decreased, citing unavailability of cash, an unstable economy and competition among traders as the main reasons for the decline. In both Eswatini and Malawi, more than half of the respondents felt that the number of traders had increased in the past 6–7 years (Table 4).

In all the study countries, a considerable proportion of craft producers sold other items including agricultural products such as fruits and vegetables, floor polish, clothes and other household items on their stalls besides crafts, with more crafters in Eswatini (41%) than in Zimbabwe (33%) and Malawi (22%) reporting doing so.

### 3.6. Markets and Income from Plant Fibre Crafts

The most important market across the three countries was passers-by (on roadsides), followed by bulk buyers, local villagers and tourists. In Eswatini, the main customers were bulk buyers (69%); while passers-by (16%), local people in the village (15%) and tourists (13%) were the other market routes. In contrast, in Malawi, crafts were sold mainly to passers-by (70%), followed by local people in the village (51%), bulk buyers (37%) and tourists (23%). Unlike in Eswatini and Malawi, the main buyers of crafts in Zimbabwe were local village members and passers-by cited by 66% and 64% of the respondents, respectively. Bulk buyers (37%) and tourists (10%) were important market routes but were mentioned by fewer respondents. A large proportion of crafters (78%) had access to formal markets in Eswatini compared to Malawi (38%) and Zimbabwe (32%) ($\chi^2$ = 80.160, *p* < 0.01).

The average gross monthly income from craft sales was modest and of the same order of magnitude across the three countries but 50% higher in Zimbabwe (US$75 ± 135) than in Eswatini (US$56 ± 71) (E785) and Malawi (US$48 ± 168) (K32,414) (F = 0.89, *p* > 0.05). However, most crafters emphasised that income was variable from month to month, with a combination of "bad months" with very little income and "good months" with high income. Most crafters in the three countries earned craft income of less than US$100 per month (Figure 3). Only a handful of crafters earned income of more than US$300 per month (Figure 3), with this being greatest in Zimbabwe (4%). Direct analysis showed that crafters who reported high income in Malawi produced large products, mainly household furniture such as cane tables, beds, couches and chairs, while those in Zimbabwe tended to produce more crafts.

Further direct analysis showed that monthly craft incomes were variable within countries. In Eswatini, there was no significant difference in craft income between middleveld sites (US$60 ± 81) and the wet, cool and mountainous highveld areas (US$45 ± 35). However, crafters with access to formal markets (US$60 ± 74) showed significantly higher craft income than those without (US$30 ± 37) (Z = 2.18, *p* < 0.05). In Malawi, crafters in the medium rainfall sites (800–1000 mm) reported greater

craft income (US$56 ± 186) than those in high rainfall sites (1200–2200 mm) zone (US$13 ± 14), although the difference was insignificant. Further, there was no significant difference in mean incomes between crafters with (US$21 ± 33) and without (US$57 ± 194) access to formal markets. In Zimbabwe, crafters in high rainfall sites (Honde) earned significantly more income (US$122 ± 206) than those in hot and dry sites, Nyanyadzi (US$33 ± 34) and Mahenye (US$62 ± 73) (F = 3.94, *p* < 0.05). Crafters with access to formal markets reported significantly higher mean incomes (US$155 ± 236) than those without access (US$49 ± 60) (*t* = 3.67, *p* < 0.001). Across all countries, there was no significant difference in mean incomes between males (US$78 ± 109) and females (US$49 ± 64) and between full-time crafters (US$73 ± 159) and part-time crafters (US$65 ± 125).

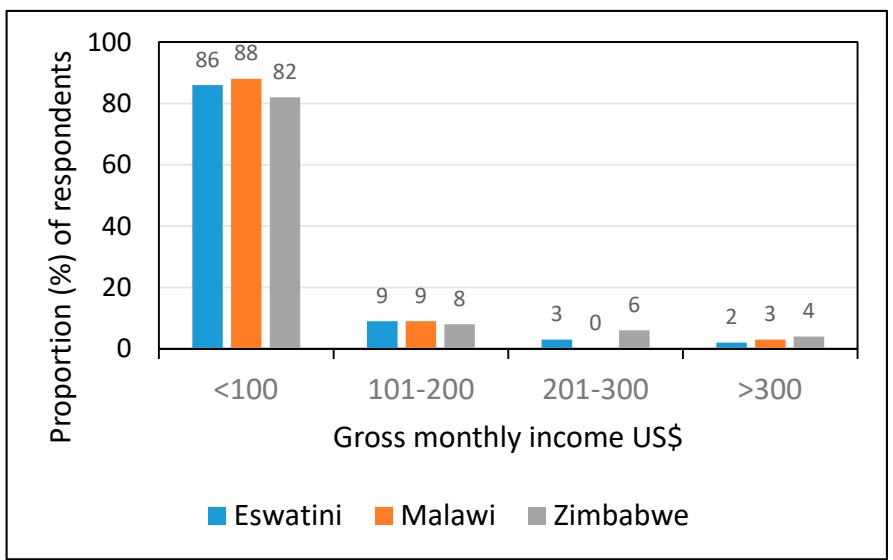

**Figure 3.** Monthly gross income from fibre craft trade in Eswatini, Malawi and Zimbabwe.

*3.7. Factors Influencing Craft Income*

Table 5 displays the results of the generalised linear regression (GLZ) model analysis performed to investigate the effects of various explanatory factors on craft income. The number of rooms in a house, period in craft business, quantity of products and variety and type of buyers showed statistically significant positive relationships with craft income, whereas age and education of crafter, household size and number of income sources yielded negative relationships.

The most common constraint to plant fibre craft trade mentioned by the respondents was the lack of infrastructure (market stalls) and poor access to formal markets for selling their products. In Eswatini, the respondents complained that they did not have adequate or enough market stalls despite that they pay a monthly fee of about E30 (US$2). Many of them sell their crafts on the market floor. Similarly, in Zimbabwe, the respondents complained that government policies around payment requirements for permits to sell at formal markets was a key problem, citing that with fewer customers they could not make profits. In Zimbabwe, the respondents also mentioned they had to travel long distances to areas where there were more customers, but this meant they incurred high transport and lodging costs. Those who sold crafts along the roadsides said they could not do so during the rainy season because (i) it was uncomfortable for them and customers to stand in the rain and (ii) some of their crafts would discolour in wet conditions. Another commonly cited problem was the scarcity of raw materials needed for craft production and, consequently, the long distances travelled. In Zimbabwe, some crafters had to travel to neighbouring Mozambique to harvest the resources, but this often resulted in conflicts with local residents of those areas. Across the three countries, the proportion of respondents citing support from government or external agencies was notably low, particularly in Malawi (Table 6).

**Table 5.** Determinants of craft income in Eswatini, Malawi and Zimbabwe, 2016.

| Variable | Coef. Estimate | Std. Error | Wald Stat. | p |
|---|---|---|---|---|
| Intercept | 3.953 | 0.603 | 42.0009 | 0.001 |
| Age of crafter | −0.035 | 0.012 | 8.544 | 0.003 |
| Gender (1 = male; 0 = female) | −0.089 | 0.134 | 0.436 | 0.509 |
| Education of crafter | −0.071 | 0.034 | 4.250 | 0.039 |
| Household size | −0.196 | 0.040 | 23.862 | 0.001 |
| Number of income sources | −1.694 | 0.248 | 46.665 | 0.001 |
| Number of rooms in house | 0.448 | 0.133 | 11.334 | 0.001 |
| Period in craft business | 0.053 | 0.013 | 17.696 | 0.001 |
| Primary plant material used (1 = from a tree; 0 = not a tree) | 0.360 | 0.294 | 1.496 | 0.221 |
| Variety of product types | 0.055 | 0.070 | 0.627 | 0.428 |
| Quantity of products | 0.014 | 0.002 | 75.884 | 0.001 |
| Variety of buyers | 0.621 | 0.201 | 9.578 | 0.001 |
| Primary buyer (1 = bulk; 0 = other) | 0.513 | 0.160 | 10.301 | 0.001 |
| Average rainfall | −0.001 | 0.001 | 3.241 | 0.072 |

**Table 6.** Proportion (%) of respondents citing support in craft production and trade.

| Support Institution | Country | | |
|---|---|---|---|
| | Eswatini | Malawi | Zimbabwe |
| Government | 10 | 3 | 32 |
| External agency | 29 | 6 | 14 |
| Local municipality | 27 | 2 | 1 |

The respondents in all the three countries said they could optimise incomes from crafts trade if (i) they had better market stalls and access to formal markets, (ii) they had access to financial support to buy inputs and equipment for craft production, (iii) they were trained to respond better to customer needs and (iv) wildfires were controlled.

## 4. Discussion

Our findings highlight both the diversity and the importance of craft production to crafters' livelihoods in developing country contexts. Moreover, several nuanced trends emerge when intercountry and intrasite participation in craft production, types and quantity of crafts produced, trade and income are considered.

### 4.1. Participation in Craft Production

Across the three countries as a whole, female participation in craft production was higher than that of men, but this was the effect of Eswatini where nearly all (99%) crafters were female. Female crafters in Malawi and Zimbabwe were in the minority. The involvement of women in high-value, resource-based activities such as craft production (wood carving) has been traditionally low in Malawi, and women tend to be mostly involved in low-value activities including fuelwood harvesting and vending forest products [26,33]. The low participation of women in Malawi and Zimbabwe may be attributed to the distance to resource sites. Where sites are far away, females might not have enough time to get there due to other household chores, whereas men can be away as long as they want and can access more distant sites [33]. The high participation of women in Eswatini can be explained by support provision through self-help arrangements and non-profit organisations such as "Gone Rural" and "Tintsaba". Evidence suggests that supporting women to venture into organised business entities can foster linkages with more lucrative markets, establishment of networking opportunities, sharing of ideas regarding consumer preferences and market trends and cushion them from costs associated with middlepersons, which can translate into better craft incomes as seen in many countries [8],

including Eswatini [37] and India [38]. Malawi had the lowest mean incomes and lowest levels of external support.

Concerning reasons for participation in craft production, our findings lend support to the importance of craft production, similar to other NTFP use activities [29] as a key livelihood source for generating or supplementing income where there are few employment opportunities and people have low education levels and lack formal skills. In our case, many households in Malawi and Zimbabwe were dependent on a diverse portfolio of livelihood sources but agriculture (crop and livestock production) was key, suggesting craft production was a supplementary income source. This is different to South Africa, where NTFP income is typically more than that from agriculture [39]. Diversification of income sources including craft production, appeared not to be a choice but a survival strategy for difficult economic situations [40]. However, it is injudicious to generalise because although mean income might be low, some households earned well above the mean. Indeed, it has been argued [19,33] that mean income per month or per year is a poor measure for NTFPs because it does not consider the amount of time or effort put into earning that income. People engage in craft trade for many reasons and to different degrees. For some, it is just a casual activity engaged in for a few hours now and again, whereas for others, it is their primary occupation and source of cash income. Thus, income should be expressed per hour worked rather than per month or per year.

### 4.2. Production of Crafts and Income

Notable trends emerge when production of crafts and income were considered by country and regions within countries. First, variations in the quantity and types of products between countries were observed, with Zimbabwe showing more product types than Eswatini and Malawi, whereas Malawi showed more quantity of crafts produced per crafter than the other two countries. Second, Eswatini had access to bulk buyers through formal markets while Malawi and Zimbabwe relied more on passers-by. Third, analysis by site shows more quantity and product types in wet regions than in dry regions for Malawi and Zimbabwe but not in Eswatini. Fourth, there was quite a lot of commonality in terms of the constraints experienced across the three countries. Taken together, the results suggest that some policy issues related to addressing constraints can be more uniform and potentially transferrable across the region but given country-specific heterogeneity, local level interventions such as transport provision and market support for supporting craft production are needed.

The mean income in the three countries (US$56–75 per month) represents about 22% of the recommended monthly minimum wage income (E3 500) in Eswatini [41] and 6% and 13% of the national monthly poverty thresholds in Malawi (MK85 852 per capita) [42] and Zimbabwe (US$562 per household) [43], respectively. The high contribution of craft income in Eswatini could perhaps be attributed to formal market access as supported by NGOs. The mean monthly income from crafts is higher than that the US$12 [12] and lower than the US$116, representing 35% of household income [13] found for different situations and products in South Africa. The low incomes can partly be explained by the fact that a very small proportion of crafters (20%) did so on a full-time basis but they also had other income sources including selling other products. Although the findings suggest that craft income represents a small proportion of the national income minimum wage levels, the levels of poverty in Malawi (51.5%) and Zimbabwe (70%) [44] are high with these figures being higher in rural areas. This means that crafting is no worse an option than many other sectors. Despite low values, the findings support the notion that plant-based craft production is likely to be an important source of income for crafter households with limited livelihood options, consistent with studies elsewhere [2,11,13,14].

These variations are possibly explained by a number and combination of factors including physical/sociocultural contexts, plant materials used, resource availability and market access. Crafters in regions with high rainfall showed significantly higher income than those in dry regions in Zimbabwe perhaps due to a greater abundance of natural resources, although a similar pattern was not evident in Eswatini and Malawi. In the dry regions of Mahenye, Zimbabwe, crafters competed for reeds with wildlife (mainly elephants) since the region borders an unfenced national park boundary. Variations in

the type and quantity of craft products produced may also be attributed to the type of plant fibre used. Different plant fibres required vary in the time required for harvesting, processing and making the crafts. For instance, the production time for bamboo-based crafts was markedly shorter than that of baobab- and reed-based crafts, as baobab fibre and reeds require more processing time before weaving.

Further, the wet region in Zimbabwe, Honde, is renowned for high agricultural productivity, especially banana farming. Thus, crafters benefit from (i) selling baskets to vendors buying agricultural produce for resale and (ii) the availability of transport carrying agricultural produce to big markets. Many of these crafters also traded in agricultural produce such as bananas, sweet potatoes, sugar cane, avocadoes, tomatoes and tea. In contrast, crop production is very limited in the dry areas except under irrigation schemes, which benefit only a few households.

The findings also show crafters with access to formal markets in Eswatini and Zimbabwe had substantially higher craft income than those without. This is likely because access to markets allows craft producers to identify customers and their needs and preferences, and to respond to these via formulating product design and marketing strategies. In Eswatini for instance, female crafters benefit from formal market access through support from NGOs such as Gone Rural. [38] note that linking up craftswomen to market outlets in Gujarat, India, allowed them to receive a fair market price for their goods without exploitation by middlepersons.

However, since the majority of crafters felt that the availability of raw materials was declining, maintaining resource extraction within ecological limits is important given the added threat of climate change in the region [7]. Although declining availability could be due to unsustainable resource harvesting practices, it may also be due to competition with wildlife in some places, bordering parks such as Mahenye, Zimbabwe, or livestock in communal systems. In Malawi, wild resource extraction for home consumption and trade has been implicated in deforestation [45]. Thus, further studies are needed to estimate extraction rates of plant resources for crafts, regrowth and the impacts of resource extraction on the resource base, in combination with other possible drivers of change.

### 4.3. Factors Influencing Craft Income

The regression analysis yielded a number of insights important for our understanding of determinants of craft income. In general, the findings lend support to craft income being more important for crafters with bigger houses, who produced more crafts, who had a long experience in craft production and sold to different types of buyers. Assets are important in shaping people choices for livelihood strategies [40]. Assets such as houses indicate wealth, e.g., [46,47], which can enable people's ability to tap into craft opportunities. A long experience in craft production yielded high craft income because of the possibility of gaining the needed craft weaving skills and knowledge of customer preferences, which could mean more products produced and traded. Some craft production is technically demanding and highly dependent on social interactions [48] and requires good knowledge of methods of production from experienced individuals mainly through verbal training, hands-on-training and observations, which can be gained over time as reflected in the findings. Selling crafts to different types of buyers can yield more income due to a big customer base. The findings show that crafters who sold their crafts to multiple and bulk buyers via formal markets earned more income than those who did not. Selling to bulk buyers can lower transaction costs (time, transport and accommodation) than selling small quantities of crafts to individual buyers.

The findings also show that craft income was lower for households with several income sources, with older crafters, with high education levels and with large households—consistent with trends with forest income [29]. A diversity of income sources did not actually translate into high income but was rather "diversification for survival," often indicative of poor welfare levels (poverty and lack of assets) [40]. Poor households made ends meet through an assortment of incomes sources including petty trade, remittances and social grants (in the case of Eswatini), and appeared not to have the assets to tap from craft income. Older household heads were likely to tap less income from crafts than younger ones, perhaps because they had acquired sufficient assets and are less dependent on

natural resources or physically less able to access resources, consistent with studies on wild resource income [17,29,49,50]. Given high poverty level in the three countries studied, the latter explanation seems more plausible than the former. Additionally, nearly two-thirds (64%) of crafters across all countries perceived that the resource availability had decreased over time, resulting in them having to walk long distances to harvesting sites. Households with high education levels tended to have low craft income perhaps due to the availability of better-paying opportunities given high education increases the chances of getting employed. Large households yielded low craft income possibly because children made up half of household members, which means they were either school-going or too young to provide labour for craft production.

## 5. Conclusions

This study represents an important step in understanding plant-based craft dynamics and income by examining craft production, trade and income across three countries and different regions within them. While the contribution of craft income relative to poverty thresholds is relatively low, except for Eswatini, the findings underscore the importance of craft production to rural livelihoods, particularly in contexts where many households are poor and struggle to make a living. Access to markets appear to enable more cash income generation from crafts trade. A nuanced contribution of our research to the understanding of crafting and rural livelihoods is highlighting that heterogeneity is significant between and within countries. Taken together, not accounting for the contribution of craft income to local livelihoods and the variability of this income would result in designing of support interventions that may not yield more craft incomes. For example, supporting local crafters with access to formal markets without (i) supporting crafters to understand customer needs for product design and (ii) addressing factors that constrain productivity (e.g., a declining resource base and transport challenges) might not result in increased incomes. Therefore, we argue for the importance of shifting the focus of craft production and income research from general macro-level studies to understanding of the more local level and design policies at that level. We suggest the following recommendations for policy focus:

- First, supporting crafters, with formal market access, should be an important consideration as it can allow identification of customer preferences and market trends and open up opportunities for increased sales.
- Second, policies for craft production and trade support should not only be macro-based but also micro-based to allow place specific-interventions as contextual realities tend to differ. For example, crafters in the remote area of Mahenye in Zimbabwe would require more support (transport and formal market access arrangements) relative to those in Honde, who benefit from agricultural markets.
- Third and last, given the high proportion of crafters who perceived that the resources are declining due to, among other things, an increase in number of crafters, strategies to better manage and even augment resources (such as through cultivation) need to be considered.

**Author Contributions:** Conceptualization, C.M.S., G.T. and D.P.; methodology, G.T. and D.P.; Software, G.T.; validation, G.T. and D.P.; formal analysis, G.T. and D.P.; investigation, G.T. and D.P.; resources, C.M.S., G.T.; data curation, G.T.; writing—original draft preparation, G.T.; writing—review and editing, G.T., D.P. and C.M.S.; visualization, G.T.; funding acquisition, G.T. and C.M.S. All authors have read and agreed to the published version of the manuscript.

**Funding:** This research was funded by Rhodes University Research Committee Grant (2017) and the South African Research Chairs Initiative of the Department of Science and Technology and the National Research Foundation of South Africa (grant no. 84379). The APC was funded by Rhodes University.

**Acknowledgments:** All community members who participated in the study and research assistants who conducted the study are thanked. G.T. is grateful to the Rhodes University Research Committee Grant for funding the study. Any opinion, finding, conclusion or recommendation expressed in this material are that of the authors and the National Research Foundation does not accept any liability in this regard.

**Conflicts of Interest:** The authors declare no conflict of interest

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
