# Peer review of "Plant Fibre Crafts Production, Trade and Income in Eswatini, Malawi and Zimbabwe"

_forests, doi:10.3390/f11080832_

Round 1

Reviewer 1 Report

The manuscript “Fibre-products production, trade and income in Eswatini, Malawi and Zimbabwe” is organized in a proper way. In general, the paper is logically structured and the overall line of the argumentation is clear. The study is providing new information on fiber products in African countries. The research questions are clear and the results answered the research questions.

This study is extremely relevant, as the understanding of dependence and livelihood contribution of plant fibre products in three African countries, as a basis for identifying strategies for optimizing benefits from production and trade of plant fibre products. There is a clear gap of knowledge of use, trade and production of fibre products, and its contribution to livelihoods, especially in African case studies. 

There are some mistakes and some issues that should be addressed:

  1. Fix the references. There are different styles of references. The journal name should be Italic (there are 3 such mistakes).
  2. There are clear four research questions and general objectives, but what about the aim? Could you please better formulate the aim of the study?
  3. In terms of the broader framework of the NTFPs, the fibre products need to be discussed, as most of NTFPs studies focus on wild food and medicine.
  4. In terms of relation to forestry and governance of forest products (or meadows or savanna or wild products in your case studies), in the Methods part please shortly describe the access to the resources (e.g. paid, free for everyone or restricted).
  5. In the introduction, the description of the use of fibre as an economic factor (safety net and income generation), influences on well-being and cultural importance. What about the results, any evidence of the use of fibre products for cultural purposes, for ceremonies and rituals or it is purely economic-driven use?
  6. It would be good to add in the discussion how nowadays the global pandemic and its economic circumstances affect or will affect the fibre products trade and the producers.
  7. The access to the market – is it the same in all study areas (there is a short description on page 13 in Constrains section). It would be good to address it in the conclusions as well.
  8. The knowledge transmission pattern, did you asked where they learned to do handicraft fibre products?
  9. On page 9 Agave is written in different sizes of the text, is it for purpose or mistake?
  10. On page 10, table 4 the Abundance of natural resources is in bold, is it for purpose or mistake? The data in table 4 could be present as a figure (which is easy to understand).
  11. It would be nice to have at least one picture with the diversity of fibre products or ideally one per each country to show the difference.
  12. Policy recommendations in conclusions should be somehow marked (use the bullet points or put it Bold), so the decision-makers see the most important message clearly.

Author Response

REVIEWER 1 RESPONSES

Comment 1

The manuscript “Fibre-products production, trade and income in Eswatini, Malawi and Zimbabwe” is organized in a proper way. In general, the paper is logically structured and the overall line of the argumentation is clear. The study is providing new information on fiber products in African countries. The research questions are clear and the results answered the research questions. This study is extremely relevant, as the understanding of dependence and livelihood contribution of plant fibre products in three African countries, as a basis for identifying strategies for optimizing benefits from production and trade of plant fibre products. There is a clear gap of knowledge of use, trade and production of fibre products, and its contribution to livelihoods, especially in African case studies. 

Response 2: Thank you for highlighting the scholarly relevance of our work.

Comment 2

Fix the references. There are different styles of references. The journal name should be Italic (there are 3 such mistakes).

Response 2: The errors have been corrected.

Comment 3

There are clear four research questions and general objectives, but what about the aim? Could you please better formulate the aim of the study?

Response 3: We have reformulated the aim as per the suggestion.

Comment 4

In terms of the broader framework of the NTFPs, the fibre products need to be discussed, as most of NTFPs studies focus on wild food and medicine.

Response 4: A pertinent suggestion. We have incorporated this in the revised version as follows:

Arguably, most studies on NTFPs mainly focus on wild foods, medicine and building material (Angelsen et al. 2014; Njwaxu and Shackleton, 2019; Nguyen et al. 2020). Despite increasing studies on plant fibre products, most are based on one or two villages or regions (Wood, 2011; Mjoli et al., 2015; Yang et al., 2018), while some are based on personal stories of individuals (Pullanikkatil and Shackleton, 2018).

Comment 5

In terms of relation to forestry and governance of forest products (or meadows or savanna or wild products in your case studies), in the Methods part please shortly describe the access to the resources (e.g. paid, free for everyone or restricted).

Response 5: We have addressed this in the revised version of the manuscript, clearly indicating that the majority of crafters in the three countries freely access plant fibre while a few purchase it.

Comment 6:

In the introduction, the description of the use of fibre as an economic factor (safety net and income generation), influences on well-being and cultural importance. What about the results, any evidence of the use of fibre products for cultural purposes, for ceremonies and rituals or it is purely economic-driven use?

Response 6: Thank you for raising this important point. Though the use of plant fibre products for cultural purposes is important, we have indicated that the scope of this study is limited to the production and livelihood uses of fibre products.

Comment 7

It would be good to add in the discussion how nowadays the global pandemic and its economic circumstances affect or will affect the fibre products trade and the producers.

Response 8: This is a very good suggestion but we are careful not speculate on the impacts of COVID-19 in these areas due to potential differential impacts on local economies and people.

Comment 9

The access to the market – is it the same in all study areas (there is a short description on page 13 in Constrains section). It would be good to address it in the conclusions as well.

Response 9: We have addressed the issue in the conclusion, highlighting that a focus on supporting access to formal markets might yield more income for crafters but that this should be supported by other locally informed interventions. For example, building market stalls would not yield more income if the crafters did not have knowledge on customer preferences and formal market access.

Comment 10

The knowledge transmission pattern, did you asked where they learned to do handicraft fibre products?

Response 10: This issue is addressed under sub-section: Reasons for production and trade of plant fibre crafts, where explain that knowledge was transmitted through parents, neighbours or training.

Comment 11

On page 9 Agave is written in different sizes of the text, is it for purpose or mistake?

Response 11:  We have corrected the highlighted error.

Comment 12

On page 10, table 4 the Abundance of natural resources is in bold, is it for purpose or mistake? The data in table 4 could be present as a figure (which is easy to understand).

Response 12: We have corrected the highlighted error.

Comment 13

It would be nice to have at least one picture with the diversity of fibre products or ideally one per each country to show the difference.

Response 13: Pictures have been provided

Comment 14

Policy recommendations in conclusions should be somehow marked (use the bullet points or put it Bold), so the decision-makers see the most important message clearly.

Comment 15: Recommendations are now presented in the form of bullet points.

Reviewer 2 Report

 Fibre-products production, trade and income in Eswatini, Malawi and Zimbabwe – short synopsis of review

The Abstract and Introduction are well-framed and explained, emphasizing the need from comparative studies.

The Methods are clear and the sample size is robust. Ethical concerns have been taken into consideration and addressed.

A few specific comments:

  • Page 8 (top): Does “need for income” have to be linked to “poverty”? I need income, and I’m not poor.

  • If possible, I would include a few illustrations in the paper.

  • The detailed quantitative information in the first paragraph of page 11 could be summarized in a table. Then the results could be interpreted and discussed in a paragraph or two. I would highly encourage the authors to explore to a greater degree the variability around the mean values (the limitations of placing too much emphasis on mean values is alluded to in the text).

  • The implications of decreased markets (buyers) is quite different depending on the underlying cause – competition versus decreased availability of raw materials. This is pointed out but could be explored to a greater degree.

  • The term formal markets is introduced at the bottom of page 11, but what “formal” market refers to is not indicated in the paper.

  • Once more, I would encourage the authors to discuss/explore variability around mean values and how this variability might be interpreted. Would it be possible to take a careful look at outliers. What aspects characterize top performers? How do they contrast from poor performers? Would it be possible to look into this?

  • Again, indicate more clearly what is meant by “formal” markets and access to them.

  • Lack of statistically significant differences is often due to high variability, but differences/trends are often apparent (for example, income earned by males versus females – see bottom of page 12). Would it be acceptable to comment on these apparent trends?

  • The authors comment on permit requirements on page 13, but provide no information on what these requirements are.

  • Reference is made to training, without any information on what this training involved – see page 14.

  • Throughout the discussion, I would try to:

  • Minimize speculative assertions – some are marked
  • Consider carefully the logic of the assertions made. For example, rather than primary plant material it would seem that the driver is final product – defines required resources and associated processing time etc. – see second to last paragraph on page 15 (highlighted)

  • Issue of mixing different topics (see 4.3 Determinants of craft income), why not develop a model or framework illustrating factors that favor and limit crafts as a livelihood strategy, based on your findings. Once you have your model, you could discuss where different contextual situations included in your study fit in.

  • The importance of designing locally relevant policies and support interventions is stressed in a couple places, but the authors do not provide any insights on what these policies and interventions might look like or how they would be developed and implemented.

  • New topics are introduced towards the end of the paper like fair trade, without indications on how this would play out.

Overall, I am confident that the authors have enough to work with to develop a good paper. I believe taking into consideration the suggestions above would help them in this effort.

Author Response

REVIEWER 2 RESPONSES

Comment 1

Fibre-products production, trade and income in Eswatini, Malawi and Zimbabwe – short synopsis of review. The Abstract and Introduction are well-framed and explained, emphasizing the need from comparative studies. The Methods are clear and the sample size is robust. Ethical concerns have been taken into consideration and addressed.

Response 1: Thank you very much for the positive feedback.

Comment 2

Page 8 (top): Does “need for income” have to be linked to “poverty”? I need income, and I’m not poor.

Response 2: Thanks for highlighting this - we have addressed this issue in the revised version as follows: “The most stated reason for entering craft production and trade was the ‘need for cash income generation’ .

Comment 3:

If possible, I would include a few illustrations in the paper.

Response 3: We interpret the request to be that the reviewer wants some photos in the manuscript. We have provided illustrations. Please see response 13, Reviewer 1. Comment 4

The detailed quantitative information in the first paragraph of page 11 could be summarized in a table. Then the results could be interpreted and discussed in a paragraph or two. I would highly encourage the authors to explore to a greater degree the variability around the mean values (the limitations of placing too much emphasis on mean values is alluded to in the text).

Response 4: This is a good suggestion but the variability of variables between and within countries makes it difficult to use a table for presentation of data. For example, there are two sites in Eswatini and Malawi but three in Zimbabwe.

Comment 5

The implications of decreased markets (buyers) is quite different depending on the underlying cause – competition versus decreased availability of raw materials. This is pointed out but could be explored to a greater degree.

Response 5: Thank you for this. We did not collect information beyond perceptions on decreased markets.

Comment 6

The term formal markets is introduced at the bottom of page 11, but what “formal” market refers to is not indicated in the paper.

Response 6: We have revised this by including the definition of a formal market as a situation where sellers can publicly advertise their prices and locations and are subject to local taxes and regulations.

Comment 7

Once more, I would encourage the authors to discuss/explore variability around mean values and how this variability might be interpreted. Would it be possible to take a careful look at outliers. What aspects characterize top performers? How do they contrast from poor performers? Would it be possible to look into this?

Response 7: The variability in means is explored later in the manuscript (as acknowledged by the reviewer on page 14. We have provided an explanation for outliers (crafters with high income) as follows:

Direct analysis showed that crafters who reported high income produced large products mainly household furniture such as cane tables, beds, couches and chairs, while those in Zimbabwe and tended to produce more crafts.

Comment 8

Again, indicate more clearly what is meant by “formal” markets and access to them.

Response 8:  See response 6.

Comment 9

Lack of statistically significant differences is often due to high variability, but differences/trends are often apparent (for example, income earned by males versus females – see bottom of page 12). Would it be acceptable to comment on these apparent trends?

Response 9: We have attempted to address this variability in the regression model.

Comment 10

The authors comment on permit requirements on page 13, but provide no information on what these requirements are.

Response 10: Thank you for picking this up. It has been addressed in the revised version as follows:

In all the three countries, most crafters had free access to resources within their localities except for a few crafters in Zimbabwe who either bought or harvested from neighbouring Mozambique.

Comment 11

Reference is made to training, without any information on what this training involved – see page 14.

Response 11: We have addressed this point in the revised version as follows:

The respondents in all the three countries said they could optimise incomes from crafts trade if (i) they had better market stalls and access to formal markets, (ii) they had access to financial support to buy inputs and equipment for craft production, (iii) they were trained to respond better to customer needs, and (iv) wildfires were controlled.

Comment 12

Throughout the discussion, I would try to: Minimize speculative assertions – some are marked; Consider carefully the logic of the assertions made. For example, rather than primary plant material it would seem that the driver is final product – defines required resources and associated processing time etc. – see second to last paragraph on page 15 (highlighted)

Response 12: We have addressed this in the revised version as follows:

Variations in the type and quantity of craft products produced may also be attributed to the type of plant fibre used. Different plant fibres require vary in the time required for harvesting, processing and making the crafts. For instance, the production time for bamboo-based crafts was markedly shorter than that of baobab- and reed-based crafts, as baobab fibre and reeds require more processing time before weaving.

Comment 13

Issue of mixing different topics (see 4.3 Determinants of craft income), why not develop a model or framework illustrating factors that favor and limit crafts as a livelihood strategy, based on your findings. Once you have your model, you could discuss where different contextual situations included in your study fit in.

Response 13: This is an important point. We have revised the sub-heading (in the results section) to factors influencing craft income. We have also integrated the sub-heading constraints to craft production with the subheading factors influencing craft income.

Comment 14

The importance of designing locally relevant policies and support interventions is stressed in a couple places, but the authors do not provide any insights on what these policies and interventions might look like or how they would be developed and implemented.

Response 14:  We have addressed this issue in the revised version as follows:

Taken together, not accounting for the contribution of craft income to local livelihoods and the variability of this income would result in designing of support interventions that may not yield more craft incomes. For example, supporting local crafters with access to formal markets without (i) supporting crafters to understand customer needs for product design and (ii) addressing factors that constrain productivity (e.g. a declining resource base) and transport challenges might not result in increased incomes.  Therefore, we argue for the importance of shifting the focus of craft production and income research from general macro-level studies to understanding of the more local level and design policies at that level. We suggest the following recommendations for policy focus:

First, supporting crafters with formal market access should be an important consideration as it can allow identification of customer preferences and market trends and open up opportunities for increased sales.

Comment 15

New topics are introduced towards the end of the paper like fair trade, without indications on how this would play out.

Response 15: We have addressed this in the revised version by removing new topics.

Comment 16: Page 11: Individual units

Response 16: The units have been provided on page 11.

Comment 17: Any insights on why these individuals were exceptional.

Response 17: We have addressed this comment – crafters who reported high incomes in Malawi mainly produced big household furniture such as beds, couches, chairs and tables.

Comment 18

Older household heads were likely tap less income from crafts than younger ones (what are the causal reasons?

Comment 18: We have addressed this comment as follows: Older household heads were likely to tap less income from crafts than younger ones, perhaps because they have acquired enough assets and are less dependent on natural resources or are less able, physically, to access resources, consistent with studies on wild resource income [17,29,49,50]. Given high poverty level in the three countries studied, the latter explanation seems more plausible than the former.

Comment 19

Overall, I am confident that the authors have enough to work with to develop a good paper. I believe taking into consideration the suggestions above would help them in this effort.

Response 19: Thank you for the comment.

Round 2

Reviewer 2 Report

I am pleased you made an effort to address the observations and suggestions. I would suggest that the entire document be read carefully to detect grammatical errors, some spelling errors and to places where a minor revision may improve clarity. 

Author Response

Thank you for the opportunity to address the minor comments highlighted by Reviewer 2.

Thank you for the opportunity to address the minor comments, as raised by the reviewer as follows:

Comment 1:  I would suggest that the entire document be read carefully to detect grammatical errors, some spelling errors and to places where a minor revision may improve clarity. 

Response 1: We have read the entire document and have made corrections where we found errors.
